# A Unified Concept-Based System for Local, Global, and Misclassification Explanations

## Abstract

Explainability of Deep Neural Networks (DNNs) has been garnering increasing attention in recent years. Of the various explainability approaches, concept-based techniques stand out for their ability to utilize human-meaningful concepts instead of focusing solely on individual pixels. However, there is a scarcity of methods that consistently provide both local and global explanations. Moreover, most of the methods have no offer to explain misclassification cases. Considering these challenges, we present a unified concept-based system for unsupervised learning of both local and global concepts. Our primary objective is to uncover the intrinsic concepts underlying each data category by training surrogate explainer networks to estimate the importance of the concepts. Our experimental results substantiated the efficacy of the discovered concepts through diverse quantitative and qualitative assessments, encompassing faithfulness, completeness, and generality. Furthermore, our approach facilitates the explanation of both accurate and erroneous predictions, rendering it a valuable tool for comprehending the characteristics of the target objects and classes.

## 1 Introduction

Understanding the behavior of intelligent models, such as Deep Neural Networks (DNNs), is becoming an increasingly important demand in the area of responsible AI Samek et al. (2019); Molnar (2020). Indeed, the models should be able to explain how, why, and when they make a particular prediction and whether their high accuracy is reliable. These requirements are necessary to establish transparency, trustworthiness, and accountability between users and models Weller (2019).

In vision applications, most explainability methods are developed using feature-based approaches, which attempt to reveal the contribution of every single feature/pixel in making certain predictions Nguyen et al. (2019). Despite the wide range of these methods proposed to date and their capabilities in addressing some application requirements, several challenges lie ahead of them. These methods usually make no effective attempt to explain the misclassification cases. They may also be fragile and vulnerable to adversarial attacks Ghorbani et al. (2019a). Moreover, these methods are inherently local, providing no perspective of what, why, and how the models generally behave Zhou et al. (2022).

These challenges have led to a surge of interest in concept-based explanation methods Achtibat et al. (2022); Mincu et al. (2021), which aim to identify the significance of a group of pixels that form a cohesive and cognitively understandable concept. Specifically, in image data, pixels are grouped into superpixels, segments, or comprehensible concepts. Although there is little consensus on these terms in the literature Schwalbe (2022), they all share a common goal of investigating the group's contribution rather than the pixels' individual roles.

The common concept learning methods typically consist of two main modules: one to form the concepts and another to score them Escalante et al. (2018). Segmentation methods are commonly used to define the concepts and scoring tools Kim et al. (2018) are widely employed to compute the concepts' importance scores. This study's contribution lies in the second module, where a recent study has shown that external scoring tools may be vulnerable methods against the perturbed input samples Uesato et al. (2018); Brown & Kvinge (2021). Additionally, they may mistakenly overestimate the concepts' importance, which makes it potentially unfaithful and sensitive to irrelevant concepts Schrouff et al. (2021). More importantly, in the supervised explanation methods, a set of prior

user-provided concept examples must be available to score and detect, while defining proper concepts and labels, and the process of labeling itself might be impossible or laborious.

These challenges prompted us to explore ways to leverage the knowledge of trained networks to eliminate the need for predefined concept examples and external scoring modules. We drew inspiration from the Outlier Exposure technique in deep anomaly detection, which uses an auxiliary dataset to teach the network better representations of anomaly cases Mirzaei et al. (2022); Hendrycks et al. (2019). Indeed, we engage surrogate explainer networks and teach them the concepts' representations of the given target against some out-of-distribution samples from unrelated classes, rather than a multi-class classification approach. This allows the networks to thoroughly learn the target's objects as well as their informative concepts.

From another perspective, similar to the feature-based explanation methods, the concepts can be extracted locally per sample Liang et al. (2021); Ancona et al. (2017), or globally per class/set of samples Ibrahim et al. (2019); Soares et al. (2021). While many methods focus on either one, few approaches could provide a consistent framework for both local and global explanations simultaneously Clough et al. (2019); Schrouff et al. (2021). For instance, Schrouff et al. (2021) combined two powerful existing techniques, one local, namely IG (Integrated Gradients), and one global, namely TCAV (Testing with Concept Activation Vectors) Kim et al. (2018). However, there is a scarcity of unified frameworks, while combining the two strategies in a single approach enhances understanding of the correct predictions and facilitates further investigation of misclassification cases. A more detailed overview of the related works can be found in the Appendix A.

In this paper, we propose a Unified Concept-Based System (UCBS) applicable to image data, which is to the best of our knowledge one of the first unsupervised unified concept extraction frameworks. This method utilizes surrogate networks and auxiliary datasets to direct the learning process towards better representations of the targets' objects as well as the targets' concepts. Indeed, a number of super-pixelated images (as the auxiliary dataset) are fed into the networks, facilitating the automatic extraction of local and global concepts of the target of interest. Having the networks adapted, the concepts are scored and the most influential ones are identified. These concept sets can serve as explanations for the network's performance. Our experimental results reveal that UCBS is able to faithfully explain different target classes with sufficient sets of concepts. Additionally, the qualitative evaluations indicate the applicability of UCBS in explaining misclassification cases and pinpointing problematic regions within an image that lead the network to incorrect predictions.

## 2 NOTATION

**Surrogate Networks** come into the scene on-demand to explain the target of interest and assist in the process of concept learning and scoring. In UCBS, we leverage pre-trained DNNs as the base models, which are adapted to obtain surrogate networks. For the case of simplicity, we call these networks $\text{Net}_\text{B}$ and $\text{Net}_\text{S}$, respectively (see Appendix B.1 for detail).

**Auxiliary Dataset** is organized from the available samples to assist surrogate networks in learning the distribution of the target concepts in addition to the target objects. Suppose that for the given target class $c$, there is a set of $n$ input images available in dataset $\mathcal{D}^\text{c} = \{x_1^c, x_2^c, ..., x_i^c, ..., x_n^c\}$. Each of these images can be segmented into $k$ segments/superpixels, i.e., for sample $x_i^c$, a set of segments as $\mathcal{S}_{x_i}^\text{c} = \{s_{i,1}^c, s_{i,2}^c, ..., s_{i,k}^c\}$ is obtained. Each segment is padded with zero value up to the original input size and labeled with the corresponding target class. To prepare the super-pixelated input data for class $c$, a segmentation method is applied to a few of its images (e.g., $m$ images), and $\mathcal{D}_\text{S}^\text{c}$ is prepared as follows:

$$\mathcal{D}_\text{S}^\text{c} = \mathcal{S}_{x_1}^\text{c} \cup \mathcal{S}_{x_2}^\text{c} \cup ... \cup \mathcal{S}_{x_m}^\text{c} \tag{1}$$

In addition, to fine-tune the binary surrogate models, we need some out-of-distribution examples, which are randomly selected from the other classes and generally contained in $\mathcal{D}_\text{out}^\text{c}$. Summing up these statements, the auxiliary dataset to explain target class $c$ is obtained as $\mathcal{D}_\text{aux}^\text{c} = \mathcal{D}^\text{c} \cup \mathcal{D}_\text{S}^\text{c} \cup \mathcal{D}_\text{out}^\text{c}$. For the sake of simplicity, we drop the index $c$ from unnecessary cases in the remainder of this paper.

## 3 PROPOSED METHOD

UCBS is an automated concept-based system that can explain a target of interest using local and global concepts. This method utilizes surrogate models and auxiliary datasets to identify the most influential

part(s) of the images and is developed using a set of super-pixelated images of the given target class, feeding the DNNs' training/fine-tuning process. The automated concept-based explanation methods are typically composed of two main stages: one to form/learn the concepts and second to score the importance of these concepts in making predictions. As Fig. 1 illustrates, the UCBS pipeline starts with training/fine-tuning surrogate models to learn and score the concepts. Then, it is followed by an extraction module, which is activated at the test time to provide local and global concepts. In what follows, we first present the algorithm for learning and scoring the concepts, then describe the concept extraction mechanism. It should be noted that while our emphasis is on fine-tuning pre-trained models to use in the post-training explanation methods, UCBS serves as a general solution that can readily be adapted for training the models from scratch.

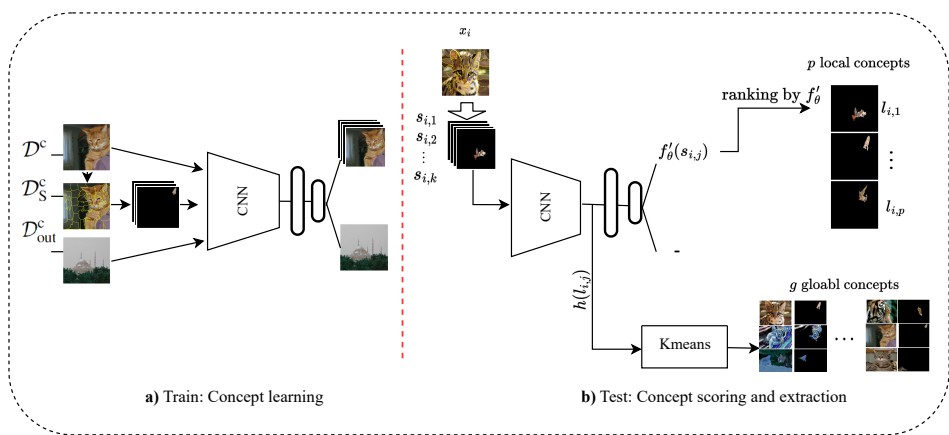

Figure 1: **UCBS general schema.** For the target of interest, (a) and (b) are successively executed; (a) surrogate networks are fine-tuned as binary classifiers (target vs. others) using auxiliary datasets including a set of super-pixelated images of the target class, the original target images, and a set of out-of-distribution samples; (b) the super-pixelated images as the potential concepts are scored and ranked using the surrogate models. To identify the concepts, the $p$ top concepts of each image are selected as the local ones, and the embeddings of a set of candidate local concepts, associated with a set of unseen images, are clustered and the $g$ top groups, indicated with their best samples, are selected as the global concepts.

**Concept learning** In the concept-based methods, particularly in image data, superpixels or segments Schwalbe (2022) are typically treated as initial concepts that their importance in making predictions should be determined and scored. Therefore, in this step, our goal is to obtain an explainer model that estimates the concepts' scores directly. To do so, we fine-tune surrogate networks, which are able to output approximate concepts' importance in a single forward pass.

Surrogate networks are indeed binary DNN classifiers built from the base (binary/multi-class) DNN models to comprehensively detect and explain samples of the given target class, i.e., the target samples vs. the non-target samples. As Fig 1-a shows, surrogate networks take the auxiliary datasets and leverage the learned representations of the base models to be fine-tuned. Given $\texttt{Net}_\texttt{S}$, in which $f_\theta : x_i \to p(\{0,1\})$ ; $x_i \in \mathcal{D}^c_{\text{aux}}$ and the original learning objective $\mathcal{L}$, we formalize the concept learning process as minimizing the following objective:

$$\mathbb{E}_{(x,y)\sim\mathcal{D}^c}\left[\mathcal{L}\left(f\left(x\right),y\right)\right] + \lambda_1\mathbb{E}_{(x,y)\sim\mathcal{D}^c_{\text{S}}}\left[\mathcal{L}\left(f\left(x\right),y\right)\right] + \lambda_2\mathbb{E}_{(x,y)\sim\mathcal{D}^c_{\text{out}}}\left[\mathcal{L}\left(f\left(x\right),y\right)\right] \tag{2}$$

in which $\lambda_1$ and $\lambda_2$ determine the relative influence of the super-pixelated and non-target samples, which are considered to be equally important in this study. This process enables the surrogate networks to learn the distribution of the concepts along with the original target images. For the zero-pixel areas of the super-pixelated images, the output of the convolution operation as well as the max or the average pooling, is zero. This prevents these regions' weights from being updated during the back-propagation operation (under some activation functions like relu or sigmoid) Apicella et al. (2021). In fact, having only the weights associated with each segment updated assists in better learning of those regions. On the other hand, this effect, together with learning the non-segmented (original) images, makes the role of the most important parts of the images highlighted. For instance,

imagine two segments in the class of tiger cat: one for a part of the ears and another for a part of the background. During the training process, on the one hand, the DNN learns these two segments are commonly associated with the tiger cat class and puts a certain concentration on them; on the other hand, in the learning of the complete images, the DNN learns that the similar parts of the former segment (ear) contribute more to identifying the tiger cat class than the background-like parts. In addition, the network observes the ear-like patterns more than the background-like data during the training process, either in the segmented or non-segmented images. The surrogate model aggregates these findings and assigns higher scores to the ear segments than to the background parts, and this is what we exactly need to discover the most important concepts.

**Concept scoring** Initial concepts must be prioritized based on their contributions to the proper predictions. To this end, UCBS utilizes the knowledge of the adapted surrogate models. Considering the aforementioned framework, the importance score of segment $s_{i,j} \in \mathcal{S}_{x_i}$, with respect to the target class $c$, is defined using $\texttt{Net}_\texttt{S}$ as follows:

$$IS(s_{i,j}) = f_\theta'^{,c}(s_{i,j}) \tag{3}$$

in which $f_\theta'^{,c}$ returns the pre-softmax output associated with the given target class. The raw output values were considered to preserve the original magnitude of the scores.

**Concept extraction** Having assigned scores to the concepts, they are ready to provide local explanations for individual data points or global explanations for either a specific target class or a set of examples. In what follows, we describe how UCBS extracts these two sets of concepts, namely local and global concepts.

**Local concepts** To extract the local concepts of an unseen image $x_i$ with respect to the class $c$, the image is first segmented and all its initial concepts are collected in $\mathcal{S}_{x_i}$, as annotated previously. The importance scores of these concepts are then obtained by passing them through the surrogate network of class $c$ and the $p$ highest-scoring ones are selected as the local concepts of sample $x_i$. These local concepts have been denoted as $l_{i,j}$ in Fig. 1-b.

$p$ is a user-defined parameter to provide different levels of complexity and accuracy, and our experiments show that $p = 3$ is a well-set value and yields the most influential parts of each image (Appendix C.2 provides an insight into the number of required concepts.). Local concepts help to deepen our understanding of FP and FN predictions, discovering why the networks misclassify input data inside or outside of the target classes (See section 4.3.3).

**Global concepts** provide a general view of each target class and reveal the common top concepts that appear in most examples of that class. To obtain these concepts, UCBS takes a set of random target images, extracts their local concepts, and groups them based on their similarity. Fig. 1-b shows this process in detail, in which first, the top local concepts of the random images are given to the surrogate network to be mapped to their embeddings ($h(l_{i,j})$). The embedding values are then clustered using their Euclidean distances as an effective perceptual similarity measure Zhang et al. (2018). The clustering results in similar concepts being grouped together as examples of a specific global concept. The clusters are sorted based on their population, and the $g$ top clusters are assigned as the final global concepts of the given target. To illustrate examples of each global concept, the densest areas of the clusters are focused. Indeed, the clusters with the highest population are more prevalent and examples with the lowest cost are more similar cases of each concept.

## 4 EXPERIMENTS AND RESULTS

This section presents a detailed discussion of the conducted experiments and evaluation of the UCBS. First, the experimental framework is described, and next, the quantitative performance of the UCBS is evaluated from different perspectives. Subsequently, we provide results of qualitative evaluations including visualization of local and global concepts, as well as the application of UCBS in explaining misclassification cases.

### 4.1 EXPERIMENTAL SETUP

In order to establish UCBS[1] as well as other comparing methods, we employed the widely-used ResNet model (34 layers) He et al. (2016), pre-trained on the ILSVRC2012 dataset (ImageNet)

---

[1]All of the code necessary to reproduce our experimental findings can be found after the reviewing process.

Russakovsky et al. (2015) as the base model. As comparisons for the UCBS scoring performance, we considered several existing approaches including two surrogate explanation methods, namely LIME Ribeiro et al. (2016) and SHAP Lundberg & Lee (2017), and two attribution methods modified for the concept-based explanations, namely GradCAM Selvaraju et al. (2017) and LRP (Layer-wise relevance propagation) Bach et al. (2015). Regarding the evaluation criteria, we used Insertion and Deletion Petsiuk et al. (2018), Sensitivity − n Ancona et al. (2017), Faithfulness Bhatt et al. (2021), and SSC and SDC Dabkowski & Gal (2017); Ghorbani et al. (2019b). Some of these measures have been adapted from the feature-based explanation approaches to the concept-based area. See Appendix B for more implementation details, evaluation metrics, baseline methods, and the applied modifications to provide concept-level results.

## 4.2 QUANTITATIVE EVALUATIONS

This section covers quantitative analysis of the provided explanations in terms of accuracy, importance, and generality. The performance of surrogate models in terms of accuracy is also evaluated in Appendix C.1.

### 4.2.1 EVALUATING EXPLANATION ACCURACY

As the proposed method is unsupervised and the true concepts and their importance are unknown, evaluating explanation accuracy is difficult. Therefore, we investigate the faithfulness of the explanations by considering Insertion and Deletion that testify to the effect of important concepts on the model predictions. To this end, we add/remove the concepts in descending order of their significance and obtain the corresponding predictions. Then the Areas Under the Insertion/Deletion Curves of the prediction probabilities are calculated (see Supp Fig. 9 as the sample curves used to calculate these results). Table. 1 summarizes the performances of various explainability methods using the average results of 1,000 random images, with respect to their corresponding true classes. The values of Insertion and Deletion show that UCBS surpassed other methods on both measures. These results indicate a higher efficacy of the UCBS method in identifying critical regions that steer the predictions toward a specific class with greater speed and accuracy. Moreover, it is noteworthy that the relative significance of the concepts is more accurately reflected in the scores assigned by the UCBS (see Appendix C.6, which visualizes score maps of different explanation methods) so that their deletion and insertion results in more desirable values of Deletion and Insertion.

In addition to the importance rankings experiments, we include results of Sensitivity − n and Faithfulness to take specifically attribution values into account. They measure the correlation of the concepts' scores and the model's predictions. The Faithfulness results in Table. 1 show that the UCBS and SHAP methods tend to be the most competitive methods of this perspective. Moreover, Supp Fig. 10 indicates the Sensitivity of various explainability methods, where UCBS performs the best overall, especially for the small subsets of concepts.

Table 1: Results of evaluating explanations accuracy as well as concepts efficiency.

|  | Insertion($\uparrow$) | Deletion($\downarrow$) | Faithfulness($\uparrow$) | SSC($\downarrow$) | SDC($\downarrow$) |
|---|---|---|---|---|---|
| LIME | 0.46 | 0.40 | 0.46 | 13.4 | **1.8** |
| SHAP | 0.64 | 0.31 | **0.65** | 9.7 | **1.1** |
| GradCAM | 0.39 | 0.33 | 0.22 | 18.8 | 2.2 |
| LRP | 0.43 | 0.40 | 0.25 | 16.6 | 2.0 |
| Ours (UCBS) | **0.71** | **0.26** | **0.66** | **8.3** | **1.6** |

### 4.2.2 EVALUATING THE EFFICIENCY OF TOP CONCEPTS

We analyze the efficiency of a concept set from two perspectives, namely completeness and compactness. The former pertains to how sufficient a particular set of concepts is in explaining the model's predictions. Compactness, meanwhile, speaks to the size of the sets, i.e., the fewer concepts, the quicker decisions. For the top concept set, these mean that a few of the most crucial concepts in the model's predictions should have the highest scores assigned. To address these aspects, we employed

and adapted two importance measures that are primarily used in pixel-based approaches; Smallest Sufficient Concepts (SSC), which determines the smallest set of concepts necessary for the accurate prediction of the target class, and Smallest Destroying Concepts (SDC), which refers to the smallest set of concepts whose removal results in incorrect predictions Ghorbani et al. (2019b); Dabkowski & Gal (2017). For a more thorough understanding of these measures, please refer to the example in Supp. Fig. 8 and the accompanying descriptions in Appendix C.2.

To assess compactness, the values of SSC and SDC (averaged over $1,000$ randomly selected validation images from the same $100$ target classes) are presented in Table. 1. In terms of SSC, the UCBS method offers the best concept set and reaches the proper predictions more quickly. However, when it comes to SDC, the methods perform similarly, typically making incorrect predictions by removing one or two of the most key concepts. On the other hand, to evaluate the completeness of the concept sets, we refer to the completeness scores, which are measured by the ratio of the accuracy attained by a given set of concepts to the original accuracy Yeh et al. (2020). Fig. 2 shows the values of completeness scores for concept sets of varying sizes. We observe that UCBS outperforms all other methods. Particularly, when selecting three concepts, UCBS achieves a completeness score above $0.5$, meaning that such a set of concepts is sufficient to attain at least $50\%$ of the original accuracy. Moreover, we draw attention to the colored circles in the figure that highlight the completeness scores of the SSC concept set, where for the UCBS approach, the SSC concept set's completeness score is $0.82$, revealing that the SSC set is highly successfully representative of the base models and by employing only $8.3$ concepts, we can accurately recover the base models' predictions up to $82\%$ accuracy.

### 4.2.3 EVALUATING THE GENERALITY OF EXTRACTED CONCEPTS

The generality of the concept-based explanations refers to how meaningful the concepts are to the other classification base models Guidotti et al. (2021). To assess this perspective of the obtained concepts, we tested the performances using various network architectures, namely Inception Szegedy et al. (2015), EfficientNet Tan & Le (2019), Vit Dosovitskiy et al. (2020), in addition to ResNet (the original base model). Indeed, the accuracy of these models is evaluated via the aforementioned process of insertion and deletion, starting with the most important concepts. Fig. 3 presents the average accuracy plots and their AUCs, where we observe consistent results between the original base model and the new ones, indicating that the extracted concepts are general yet effective in explaining different target classes irrespective of the employed model in the learning phase. We conducted statistical tests to further assess the relative performance of the base models in comparison to one another, as detailed in Appendix C.4.

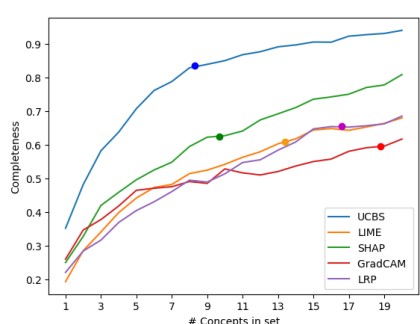

Figure 2: Completeness scores for the concept sets of varying sizes.

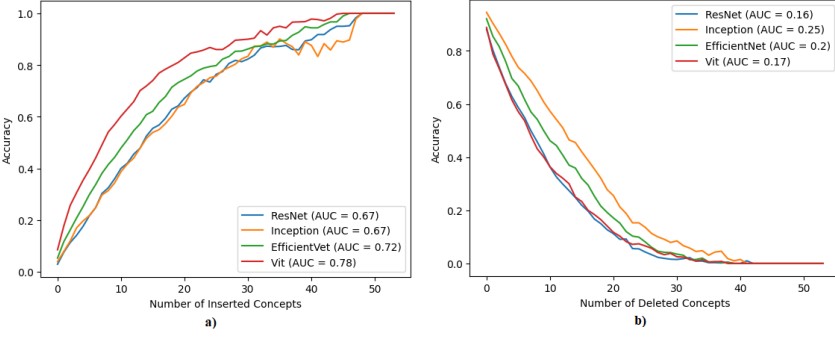

Figure 3: Evaluating the performance of the obtained concepts using different base models, a) when the concepts are progressively inserted, b) when they are deleted.

### 4.3 QUALITATIVE EVALUATIONS

In the following sections, the outcomes of local, global, and misclassification explanations of the UCBS are provided, specifically for three target classes: tiger cat, police van, and revolver. Additional qualitative examples are also available in the Appendix C.6 and C.7.

#### 4.3.1 LOCAL EXPLANATIONS

Fig. 4 displays examples of local concepts for the three classes. The first two rows suggest that (a portion of) the animals' faces and their body texture play a significant role in identifying the tiger cat objects. In the case of the police van examples, the figures illustrate that the network focuses on the police logo and various vehicle components, such as the tire, the windows, and the headlight. For the revolver, the extracted local concepts verify that the network can successfully recognize different parts of these objects, including the grip, the cylinder, and the barrel.

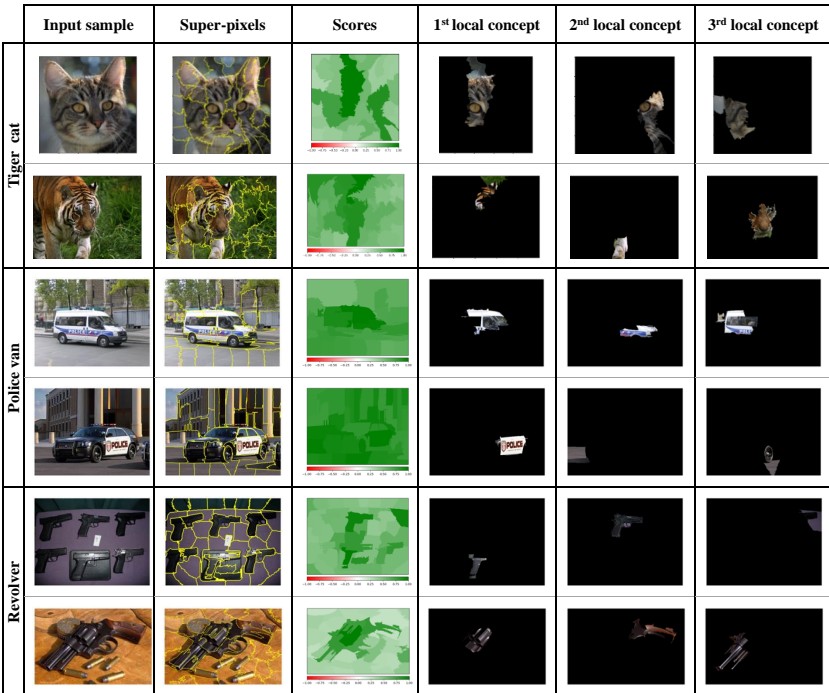

Figure 4: Local concepts to explain individual images. For each class, there are two examples, displaying the original image, the segmented version, the complete scoring map, and the top three local concepts of each instance.

#### 4.3.2 GLOBAL EXPLANATIONS

To illustrate the global concepts, UCBS was applied to the three aforementioned target classes, and Fig. 5 depicts their outputs. In this figure, the top three global concepts of each class have been indicated via the best three representative samples (having minimum cost) of each group. About the tiger cat class, UCBS proposes that the most crucial and general concepts for identifying such objects include the animal's (pointy) ears, the body texture (incorporating fur patterns), and a portion of their mouth and nose (featuring cat whiskers). For police vans, it is evident that the network has primarily concentrated on the standalone police text logo, the police text logo positioned on the front of the vehicles (encompassing a section of the car bumper and headlight), and the van's tire, respectively. Regarding the revolver, UCBS clarifies that objects of this category can effectively be recognized using the concepts of the barrel (incorporating the front sight), the cylinder, and the trigger.

#### 4.3.3 EXPLAINING MISCLASSIFICATIONS

Explaining the network's functionality based on the most impactful concepts assists in the clarification of misclassification cases, either FP or FN ones. Fig. 6 presents one example of FP and one example of FN for each target class. The first two rows pertain to the tiger cat class, where a cat image

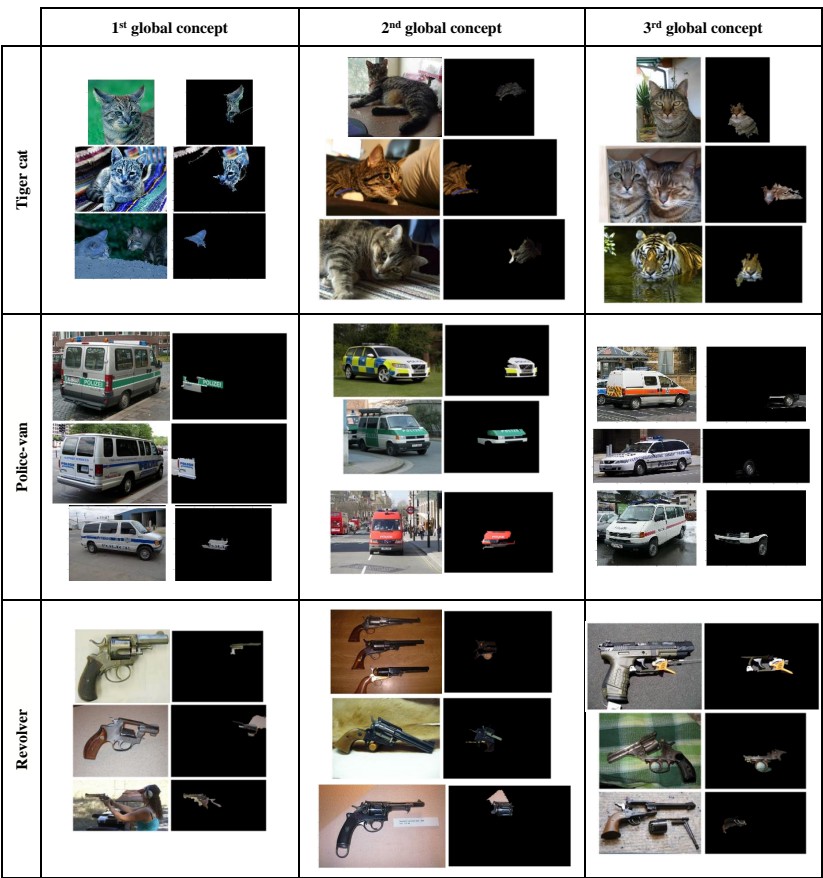

Figure 5: Global concepts to explain one certain target class. For each class, the top three global concepts are indicated via the best three representative samples of each group. The input images are shown in their original sizes.

could not be recognized and a dog image was mistakenly placed in the tiger cat class, respectively. Considering the local concepts of the former case, it becomes apparent that the network struggled to identify similarities to the global concepts of this class and primarily focused on irrelevant concepts. Conversely, the striped-shaped portions of the dog image and its front paws led the network to misclassify that image. The following two rows are associated with the police van class, where no police logo or vehicle parts were recognized in the first case, and conversely, parts of the windshield, bumper, headlight, and bus tire were detected in the FP case. These local concepts contributed to the incorrect classification of the objects. In the FN case of the revolver, the network failed to identify some influential parts of the object, as suggested by the global concepts. Additionally, in the FP case, the detected tube-like part of the object may have misled the network.

## 5 DISCUSSION, LIMITATIONS, AND FUTURE WORKS

The UCBS method is designed to automatically extract human-understandable and identifiable concepts, such as "pointy ears" and "police logo", eliminating the need for providing user-defined concepts and applying external scoring methods. Providing human-meaningful concepts is beneficial for explaining results to (non-expert) end-users, and the ability to comprehend local and global concepts in a unified framework is promising for clarifying false predictions. This method is developed based on image data, however, the general idea of taking advantage of surrogate models to provide unified concept-based explainability frameworks is applicable to other data types.

We demonstrated the effectiveness and generality of the extracted concepts. However, it is important to acknowledge that the limitations and drawbacks of superpixels are inherited by the UCBS, which may potentially result in the creation of meaningless concepts. Moreover, the automatic extraction

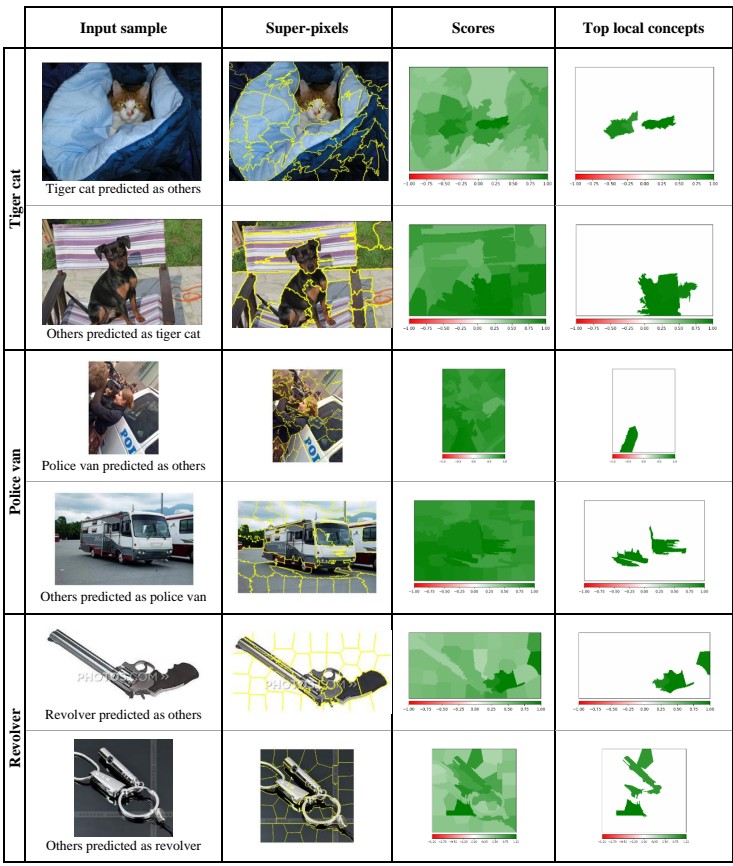

Figure 6: Explaining misclassification cases: two examples of FN and FP are represented for each target class, and the input image (in its original size), the segmented version, the complete scoring map, and the scoring map of the top three local concepts are indicated for each example.

of abstract and complex concepts may be challenging or necessitate additional post-processing. Addressing these challenges presents an intriguing direction for future research, particularly in identifying and surrounding the most influential concepts within bounding boxes, which could help mitigate the issues. In addition, in this study, we used a binary classifier for each target class in the concept learning and network adaptation process. Although only a few epochs are required to fine-tune the surrogate networks, the approach may seem time-consuming when applied to multi-class datasets. Therefore, we plan to apply super-pixelated images for the simultaneous concept learning of multiple target classes as a whole in future research. Appendix C.5 presents several examples of the limitations, including imperfectly super-pixelated inputs and dataset biases.

## 6 Conclusion

This work proposed UCBS to bridge the gap between local and global explanation techniques as well as to analyze misclassification cases. We took advantage of super-pixellating of the training data and incorporating them into the learning process. This resulted in not only improving the performances but also facilitating the identification and scoring of the most influential concepts. In practical applications, we are interested in obtaining a cognitively comprehensible rationale for the model's predictions, either for a single data point or a set of them. In the proposed UCBS, the former was achieved by ranking the concepts locally, and the latter was attained by tracking the concepts globally. Furthermore, by closely monitoring the local concepts and comparing them to the global ones, UCBS was able to clarify the false predictions. Insights gained from this research may promote the safer use of AI models in vision-based applications within safety-critical fields such as healthcare or finance. Additionally, these insights may facilitate the design, development, and debugging process of the models, from the perspective of the developers.

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

APPENDIX

## A. BACKGROUND AND RELATED WORKS

In vision-based applications, the significance of the pixels is usually determined in the scope of a single image, and it is not straightforward to determine important pixels for a variety of images of a certain class Ribeiro et al. (2016). However, the fact is that the practitioners are eager to make sense of the overall reasoning of the models in addition to the individual explanations Tonekaboni et al. (2019); Zarlenga et al. (2021). Wu et al. (2020) proposed a novel two-stage framework composed of a feature occlusion analysis and a mapping stage, which explains the model decisions in terms of the importance of category-wide concepts. Summit Hohman et al. (2019) proposed a dynamic platform to summarize and visualize the learned features of a DNN and their interaction. It introduced two summarization techniques, namely activation aggregation and neuron-influence aggregation.

Despite these single-purpose works, having local and global approaches in a single framework can make explanations practical for comparing assessment tasks, especially in the concept-based approaches where we are not faced with firm pre-defined concepts (ground truths). Indeed, in unified frameworks Lundberg et al. (2020); Schwalbe (2022); Lundberg & Lee (2017), local and global explanations can directly be contrasted. This assists in highlighting the unique characteristics of each sample against the representative concepts of the understudied target class. In this context, there are few frameworks, and most of them extract and score discovered concepts in separate modules. Ghorbani et al. proposed ACE, an Automated Concept-based Explanation, which aggregates relevant image segments across multiple input data to obtain global concepts Ghorbani et al. (2019b). ACE tends to provide global concept samples and has no straightforward suggestions for local interpretations of either true or false predictions. Moreover, these methods used TCAV to score the discovered concepts, which has its own fragility, as discussed in the introduction. It is worth mentioning that TCAV and UCBS are not directly comparable, because TCAV is a supervised explanation method and needs predefined concept examples, unlike UCBS.

## B. ADDITIONAL IMPLEMENTATION DETAILS

This section provides additional details on implementation settings, evaluation metrics, baseline comparison methods, and the applied modifications to provide concept-level results.

### B.1 BASE MODELS AND HYPER-PARAMETERS SETTINGS

We selected the ResNet model He et al. (2016), comprising 34 layers and pre-trained on the ImageNet dataset, as the base model. This network has the classification head for a total of 1000 classes. To adapt this network for binary classification (surrogate networks), we replaced the classification head with a binary one, as illustrated in Supp. Fig. 7. Furthermore, the maximum softmax probability baseline detector and the cross-entropy loss were used to fine-tune the networks.

The target classes were randomly selected among the ImageNet classes (generally 100 classes to report the following measures). For each target class, we segmented 50 random images of the training set into 50 super-pixels using the SLIC segmentation method Achanta et al. (2012). This resulted in a maximum of $2,500$ super-pixelated images. The SLIC segmentation method was employed due to its speed and simplicity. Furthermore, $1,300$ random images from the other classes were selected to make the out-of-distribution dataset, and they were used to fine-tune the surrogate models for a few epochs (50). This UCBS hyper-parameters setup proved sufficient for our experiments with the ImageNet dataset, allowing the networks to learn the targets' concepts in addition to the original targets' objects.

During the testing phase, three local concepts were extracted and visualized for each input image, and three global concepts were identified for every target class (see Appendix C.2 about the number of required concepts). The latter were obtained using the candidate local concepts of 20 unseen images. To determine the global concepts, we performed KMeans clustering with $K = 5$, and the three best representatives of each group are illustrated in the following figures.

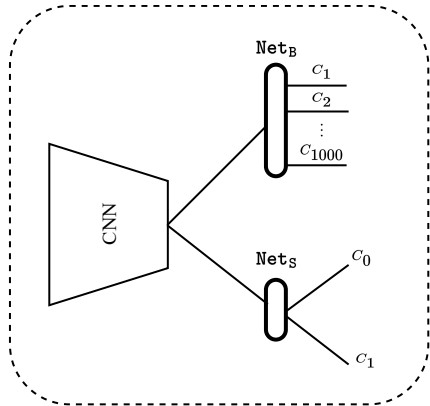

Supplementary Figure 7: The base and surrogate networks

## B.2 EVALUATION METRICS

This section presents additional details about the evaluation criteria used in the experiments of this study (Section 4).

- **Deletion** and **Insertion**: These two metrics are to investigate the faithfulness of an explanation to the base model. They explore whether pixels/concepts with high scores are really important to the predictions. They operate on the premise that the removal or insertion of the regions important to a network will force the base model to alter its decision. Specifically, the Deletion metric quantifies the reduction in the probability of the predicted class as progressively more important pixels are removed, i.e., the sharper drop and consequently lower Area Under the probability Curve (AUC) indicate strong explanations. Conversely, the Insertion metric takes a complementary approach. It gauges how fast the probability increases as pixels are inserted into an empty canvas in descending order, i.e., the larger Insertion AUC, the better faithfulness Petsiuk et al. (2018). In this study, we adapted the Deletion and Insertion AUCs for the concept-based applications, in which the deletion and insertion operations were applied using the concepts of individual images, and then the AUC results were averaged across $1,000$ random examples for their true classes. We call these metrics Deletion and Insertion for short.

  In addition to the above local measures, in order to evaluate the faithfulness of the extracted concepts to different base models, we also adapted these measures for the global evaluation of a set of examples. Indeed, we calculated the Deletion/Insertion AUCs for the accuracy curves of the validation sets obtained from different base models. In this way, we are able to evaluate how precise the concepts/explanations are, considering new base models and regardless of the model employed in the scoring phase.

- **Sensitivity** $-$ **n**: The aim of this metric is to assess whether the attribution scores are in line with the changes in the model's predictions when features are removed. To achieve this, different subsets of features are randomly selected and their individual attribution scores are computed. Then, the correlation between the predictions made with each subset and the sum of their respective attribution scores are calculated. Typically, the calculations are performed across subsets with a fixed size of $n$. This is done repeatedly for multiple predictions, and the results are averaged for a more accurate metric Ancona et al. (2017); Covert et al. (2023). We also adapted this metric for the extracted local concepts and computed it for subsets ranging from 3 to a maximum of 50 concepts, with a step size of 3. To obtain these measures, we utilized $1,000$ random images, each of which was sampled 100 times.

- **Faithfulness**: is a similar metric to Sensitivity-n in which the correlation is jointly calculated across subsets of all sizes Bhatt et al. (2021). Similarly, the Faithfulness values were obtained by calculating them for $1,000$ random images, each of which was sampled 100 times.

- **SSC** and **SDC**: These two metrics are used to evaluate the efficiency of a concept set. SSC (Smallest Sufficient Concepts) seeks the smallest set of concepts necessary for the accurate prediction of the target class, and SDC (Smallest Destroying Concepts) refer to the smallest set of

concepts whose removal will result in incorrect predictions Ghorbani et al. (2019b); Dabkowski & Gal (2017). These measures can be considered as local metrics, i.e., for input sample $x_i$, the values of $\mathrm{SSC}(x_i)$ and $\mathrm{SDC}(x_i)$ are calculated with the cardinality of the aforementioned sets obtained from $x_i$. From the global point of view, they are averaged over all the images of a dataset ($\mathcal{D}$) and notated as $\mathrm{SSC}(\mathcal{D})$ and $\mathrm{SDC}(\mathcal{D})$.

- **Completeness**: quantifies how sufficient a particular set of concepts is in explaining a model's predictions. It is calculated using the ratio of the accuracy attained by a given set of concepts to the original accuracy Yeh et al. (2020).

### B.3 BASELINE METHODS

This section provides an overview of the baseline explanation methods. We used two surrogate models, one gradient-based and one activation-based method, and we modified the last two ones to arrive at concept-level attribution scores.

- **LIME** ( Local Interpretable Model-agnostic Explanations ): is an explanation method that can be applied to any classifier or regressor in a faithful way. LIME works by perturbing the input features and locally approximating the model around the given predictions. It generates a dataset of modified samples and trains an inherently interpretable model over it. Then, the surrogate trained models are used to compute feature importances and explanations for the original model's predictions Ribeiro et al. (2016).

- **SHAP** (SHapley Additive exPlanations): is an explanation method that uses Shapley values, a concept from cooperative game theory, to assign a numerical value to each feature. Shapley values measure the contribution of each feature in predicting the outcome, considering several combinations of the features. SHAP uses a model-agnostic algorithm to estimate the contribution scores and they are aggregated across samples to generate the final features' importances Lundberg & Lee (2017).

- **GradCAM** (Gradient-weighted Class Activation Mapping): is an explanation method that highlights the regions of an input image that are important for the decision of a convolutional neural network (CNN). It works by computing the gradient of the predicted class score with respect to the feature maps of the last convolutional layer of the CNN and generates a heatmap that visually indicates which parts of an image the CNN is focusing on to make its prediction. The gradients represent the importance of each feature map in predicting the output class. This method is useful in object detection, segmentation, classification, and other computer vision problems Selvaraju et al. (2017).

- **LRP** (Layer-wise relevance propagation): is a backpropagation-based neural network explanation method that computes each input feature's contribution to the output. It computes the relevance scores of each neuron, which score represent the contribution of the neurons to the prediction. LRP works by dividing the output relevance scores proportionally to the input features that were responsible for the neuron's activation during the forward pass. This process is done layer-by-layer with the relevance scores being propagated backwards through the layers of the network Bach et al. (2015).

To tailor the GradCAM and LRP methods for the concept-based applications, we calculate the weight of each concept (superpixel) by adding up the absolute values of its constituent pixels. However, we explored alternative approaches to transform the pixel values into superpixel scores, we concluded that the sum of absolute values was the most effective one, as reported in similar works Hartley et al. (2021).

## C. ADDITIONAL RESULTS

This section provides additional experimental results.

### C.1 EVALUATING PERFORMANCE OF SURROGATE MODELS

Surrogate models are trained to imitate the behavior of the original base models, making them more straightforward to explain and analyze. Therefore, incorporating super-pixelated images in

the fine-tuning process should not compromise the performance of the models. To investigate this, we evaluated the discrimination capability of the models using the ImageNet validation set. We conducted two assessments: one without the super-pixelated samples and another with them. We examined the validation set in three different scenarios: 1) Only including images from the target class (val $_{pure}$), 2) Including an equal number of images from the target class and the other class (val $_{equal}$), and 3) Including images from the target class and randomly selecting $1,000$ images from other classes (val$_{1000}$). Supp. Table 2 presents the average loss and accuracy values for these three scenarios.

Supplementary Table 2: The performance results of Net$_S$, with and without feeding the super-pixelated samples.

| State | val $_{pure}$ | | val $_{equal}$ | | val$_{1000}$ | |
|---|---|---|---|---|---|---|
| | ACC(%) | Loss | ACC(%) | Loss | ACC(%) | Loss |
| Without | $93.16 \pm 6.09$ | 0.37 | $93.36 \pm 4.72$ | 0.37 | $92.78 \pm 4.30$ | 0.38 |
| With | $95.93 \pm 4.29$ | 0.35 | $95.04 \pm 4.19$ | 0.36 | $94.80 \pm 3.79$ | 0.36 |

The findings clearly demonstrate that the networks' discrimination capabilities have not only been maintained but also improved across all scenarios. This improvement can be attributed to the utilization of super-pixelated images, which facilitate more effective learning of the target objects. In other words, they progressively assist in focusing on different parts of the target images, compelling the networks to grasp the finer details of the target concepts in addition to the overall target objects. Consider that for a super-pixelated sample, the weights of the superpixels are updated, while the corresponding weights of the (black) padding pixels remain unchanged. Furthermore, the super-pixelated samples aid the networks in learning the common and frequent concepts associated with each target class. As a result of these two features, the networks are directed to make more precise predictions based on the importance and relevance of the target concepts rather than the role of individual pixels. This effect persists even when the number of images from other classes increases up to $1,000$ cases, implying that learning the target concepts enhances the networks' ability to better differentiate the non-target objects as well. Consequently, this approach can be considered a valuable tool for directing the networks to make predictions based on the target concepts and mitigating the influence of irrelevant factors such as background, potential biases, and so on.

## C.2 INSIGHTS INTO THE NUMBER OF REQUIRED CONCEPTS

To illustrate the **SSC** and **SDC** measures, Supp. Fig. 8 has been provided. It shows the images including/excluding the top concepts in addition to the values of SSC and SDC for one example of the tiger cat class. We passed these images through the Net$_S$ as well as Net$_B$ and reported the predictions below of each image. We did this experiment to evaluate the effectiveness of the obtained concepts in distinguishing the target images not only in a binary manner (target or non-target) but also among all the 1000 classes of the ImageNet. In the latter, the tiger cat object can wrongly be recognized as a Conch, a Tabby, an Egyptian cat, a Persian cat, and so on.

In this context, the minimum number of top local concepts to recognize objects of target class $c$ is obtained as follows:

$$p^c = Max\big( \, \text{SSC} \, (\mathcal{D}^c) \, , \, \text{SDC} \, (\mathcal{D}^c) \, \big) \tag{4}$$

in which $\mathcal{D}^c$ is the training set of this target. Although it is better to determine the values of $p^c$ per target class, we suggest defining hyper-parameter $p$ as follows:

$$p = Max \left( \frac{1}{|C|} \sum_{c \in C} \text{SSC} \, (\mathcal{D}^c) \, , \, \frac{1}{|C|} \sum_{c \in C} \text{SDC} \, (\mathcal{D}^c) \right) \tag{5}$$

in which the average values are calculated across all the considered classes and utilized in the max operator to ensure that no useful concepts are lost. These settings are applicable for both base and surrogate networks.

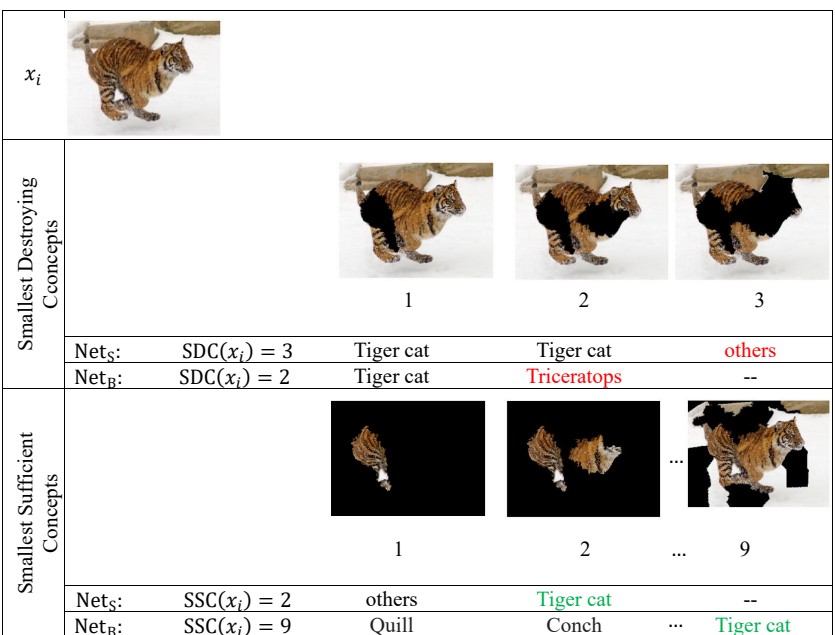

| | | | 1 | 2 | 3 |
|---|---|---|---|---|---|
| Net$_S$: | SDC($x_i$) = 3 | | Tiger cat | Tiger cat | others |
| Net$_B$: | SDC($x_i$) = 2 | | Tiger cat | Triceratops | -- |

| | | | 1 | 2 | ... | 9 |
|---|---|---|---|---|---|---|
| Net$_S$: | SSC($x_i$) = 2 | | others | Tiger cat | | -- |
| Net$_B$: | SSC($x_i$) = 9 | | Quill | Conch | ... | Tiger cat |

Supplementary Figure 8: Illustration of SSC and SDC for an instance of the tiger cat class.

According to our experiments, which involve 100 selected classes from the ImageNet dataset, we set the value of $p = 3$ to distinguish target class images from all unrelated ones. The average values of SSC and SDC are reported in Supp. Table. 3. These values indicate that Net$_S$ requires at least $3.6$ concepts to perform proper predictions. On the other hand, removing an average of $3.4$ key concepts from the images causes Net$_S$ to make inaccurate predictions. For Net$_B$, these values are $7.93$ and $1.9$, respectively.

Comparing the SSC measures of the two networks implies that explaining targets in a binary manner is more straightforward and requires fewer concepts than the multi-class approach, which necessitates a greater number of concepts (Net$_B$). To put it another way, as illustrated in Supp. Fig. 8, it takes longer for Net$_B$ to recognize a tiger cat among other specific classes, some of which resemble the tiger category closely (such as Tabby, Egyptian cat, and Persian cat), than it does for Net$_S$ to comprehend that a tiger cat is a tiger cat and nothing else in general. Furthermore, from another perspective, the SDC values demonstrate that Net$_B$ is prone to make errors more quickly than Net$_S$.

In summary, the concepts extracted using surrogate networks are useful and important, as we generally require 7 such concepts for proper recognition of an image among the 1000 classes of ImageNet images. Despite these findings, we hypothesize that integrating these networks and feeding super-pixelated images into the main network will lead to lower numbers and eliminate the need for individual fine-tuning processes, which will be investigated in our future plans.

Supplementary Table 3: The values of SSC and SDC over all the training target classes.

| | $\overline{\text{SSC}}$ | $\overline{\text{SDC}}$ | $p$ |
|---|---|---|---|
| Net$_S$ | 3.6 | 3.4 | 3 |
| Net$_B$ | 7.93 | 1.9 | 7 |

## C.3 COMPLEMENTARY RESULTS OF EVALUATING EXPLANATION ACCURACY

To complete the findings of Section 4.2.1, we include Supp Figs. 9 and 10. The former is for the local performance of the UCBS and the latter indicates the $\text{Sensitivity} - \text{n}$ of various explainability methods. In Supp Fig. 9, several examples with their concept scoring maps and the Insertion and

Deletion curves have been illustrated. These curves were utilized to compute the local AUC per sample and then average across all the random images to have the final Insertion/Deletion measures in Table. 1. As stated, the higher Insertion AUC and the lower Deletion are desirable, indicating the higher faithfulness of the concepts.

Sensitivity values were calculated using $1,000$ random images and then they were averaged across different subsets of concepts with step size 3. Supp Fig. 10 illustrates that UCBS outperforms the other methods overall, particularly when dealing with small subsets of concepts, while SHAP also demonstrates strong performance. These findings provide further evidence of the significant correlations between the concepts' importance and models' predictions, as indicated by the Faithfulness measures.

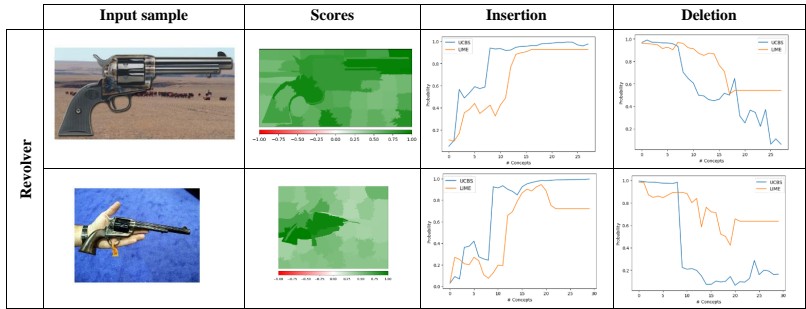

Supplementary Figure 9: The local performance of UCBS: Left) original images and the concepts' scores, Right) the Insertion and Deletion curves, which indicate the probability of the corresponding class after the insertion or deletion of important concepts (higher AUC is better for insertion, lower for deletion).

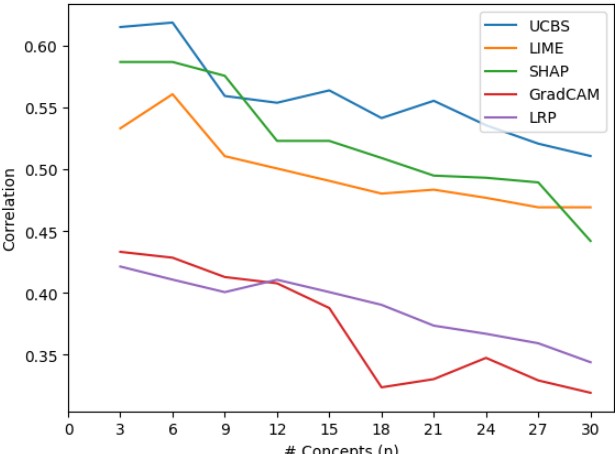

Supplementary Figure 10: Evaluation of Sensitivity $-$ n. The measures were calculated for various subsets of concepts (step size 3) and then averaged across all $1,000$ random images.

### C.4 COMPLEMENTARY RESULTS OF EVALUATING THE GENERALITY OF THE CONCEPTS: STATISTICAL TESTS

In order to calculate the Insertion/Deletion accuracies presented in Fig. 3 of section 4.2.3, we used $1,000$ random samples from the validation sets of the considered 100 target classes. Each image was processed by various base models, and the accuracy of each class was computed. Then, the accuracies were averaged to generate the final Insertion/deletion plots. Before averaging, we applied Freidman's test Friedman (1937) to statistically evaluate and rank the base models. Supp. Table .4

presents the results and its last row shows that the default equivalency hypothesis $H_0$ is rejected, because $p - value$ is lower than $\alpha$. This means that the models perform differently in this perspective. Therefore, we refer to the rank values, where the lower ranks indicate better performance. The results indicate that, for insertion, Vit and EfficentNet are the top-ranked models, which surprisingly outperform the original base model, namely ResNet. On the other hand, for deletion, ResNet and Vit are the best performers. The AUC values displayed in Fig. 3 confirm these ranking results (consider that, as noted in Appendix B.2, when the best concepts are progressively removed, the quickest performance drops and, therefore, the lower accuracies are more desirable, implying that the removed concepts are more meaningful to the corresponding base model.). Summing up, the results suggest a potential connection between the concepts' performance and the patch-based learning method used by Vit, and this model showcases the most promise for our future analyses.

Supplementary Table 4: Results of ranking the accuracy of different base models, using Freidman's test with significance level $\alpha = 0.1$.

| Models | Insert Rank($\downarrow$) | Delete Rank($\downarrow$) |
|---|---|---|
| Vit | **1.18** | 1.91 |
| EfficientNet | 2.11 | 2.75 |
| ResNet | 3.27 | **1.69** |
| Inception | 3.42 | 3.60 |
| $p - value = 1e - 5$ | $H_0$ is rejected | $H_0$ is rejected |

C.5 MORE DISCUSSION ON LIMITATIONS: IMPERFECTLY SUPER-PIXELATED INPUTS AND DATASET BIASES

Applying super-pixelating methods, even the multi-resolution ones, to obtain initial segments presents several limitations; large segments containing multiple meaningful concepts or very small segments with no discernible concept can be generated; a single concept may be fragmented across several segments. Moreover, super-pixelating methods typically offer no solution for tracking entirely or partially abstract concepts such as brightness, daylight, and sun angle in sunrise or sunset images. Supp. Fig. 11 illustrates some of these cases. Future works remain to see whether a sliding window approach that reflects the influence of the network's prediction on different parts of the images could overcome these issues.

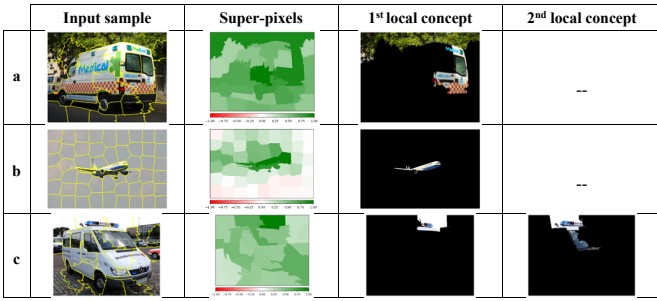

Supplementary Figure 11: The effect of imperfectly super-pixelated inputs: a) a large superpixel contains background b) Several concepts (different parts of the airplane) in one superpixel c) one concept (blue-red ambulance light) fragmented into two segments.

Our experiments encountered another challenge. That is the possible dataset biases, where the extracted concepts are considerably influenced by the images available in the dataset. This results in concepts that may deviate from human intuition. For instance, as illustrated in Supp. Fig. 15, the most important concept for predicting mountain bike images is the driver rather than the bike itself. This occurs because most ImageNet mountain bike images mainly contain drivers. Another example is the restaurant class, which contains images that differ from the main context, rendering the UCBS

concepts inapplicable and leading to unhelpful misclassification explanations, as shown in Supp. Fig. 12. Such biases present considerable challenges to explainable methods, and exclusive research is necessary for devising methods that mitigate them.

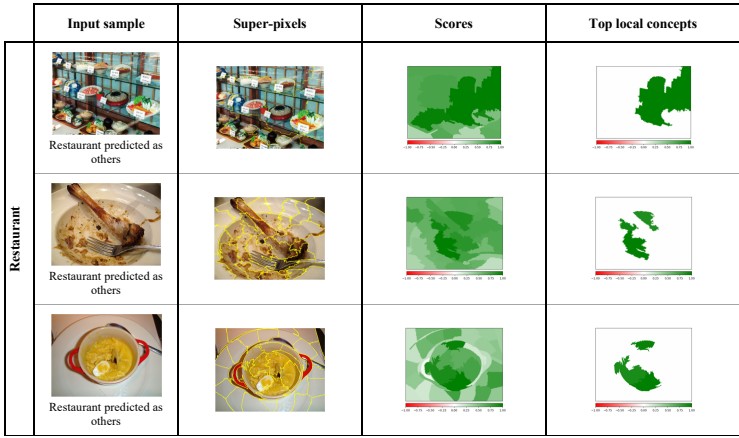

Supplementary Figure 12: Restaurant uncommon images and failure of misclassification explanations.

## C.6 MORE QUALITATIVE EXAMPLES

This section provides more qualitative examples for the UCBS as well as the baseline explanation methods. Supp. Figs. 13 shows a comparison between explanations for several examples from the ImageNet dataset. As can be seen, in UCBS, the importance scores are relative scores; even background regions have a slight influence in making predictions. This is because UCBS assigns the importance score to a concept considering all other parts of the image that the model has already learned, whereas in the other methods, concepts are investigated either completely or partially independent of each other. This may lead to some concepts being overlooked even though they indirectly influence predictions. For example, the color and texture of the basketball court, the street asphalt in detecting a police van, or a human hand holding a revolver.

## C.7 MORE EXAMPLES OF UCBS

Results of local, global, and misclassification explanations for six other ImageNet classes are provided in Supp. Figs. 14-19.

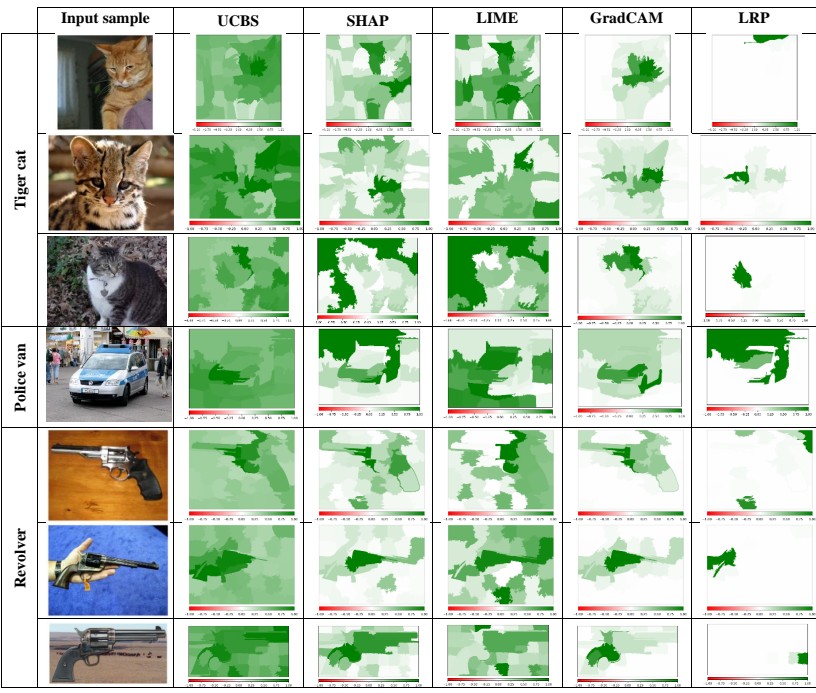

Supplementary Figure 13: More qualitative examples of the UCBS method as well as the baseline explanation methods.

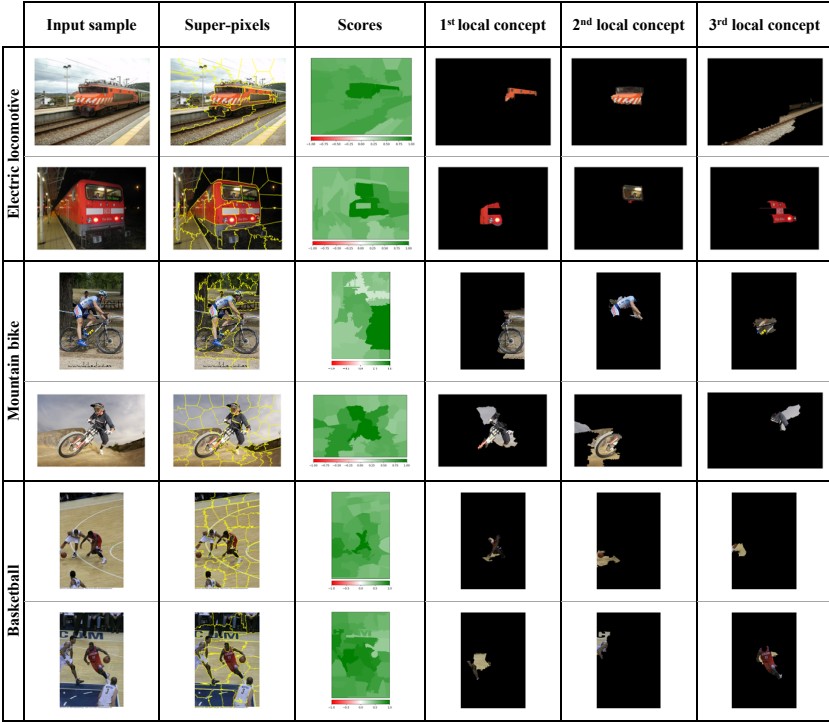

Supplementary Figure 14: Local concepts to explain individual images. For each class, there are two examples, displaying the original image, the segmented version, the complete scoring map, and the top three local concepts of each instance.

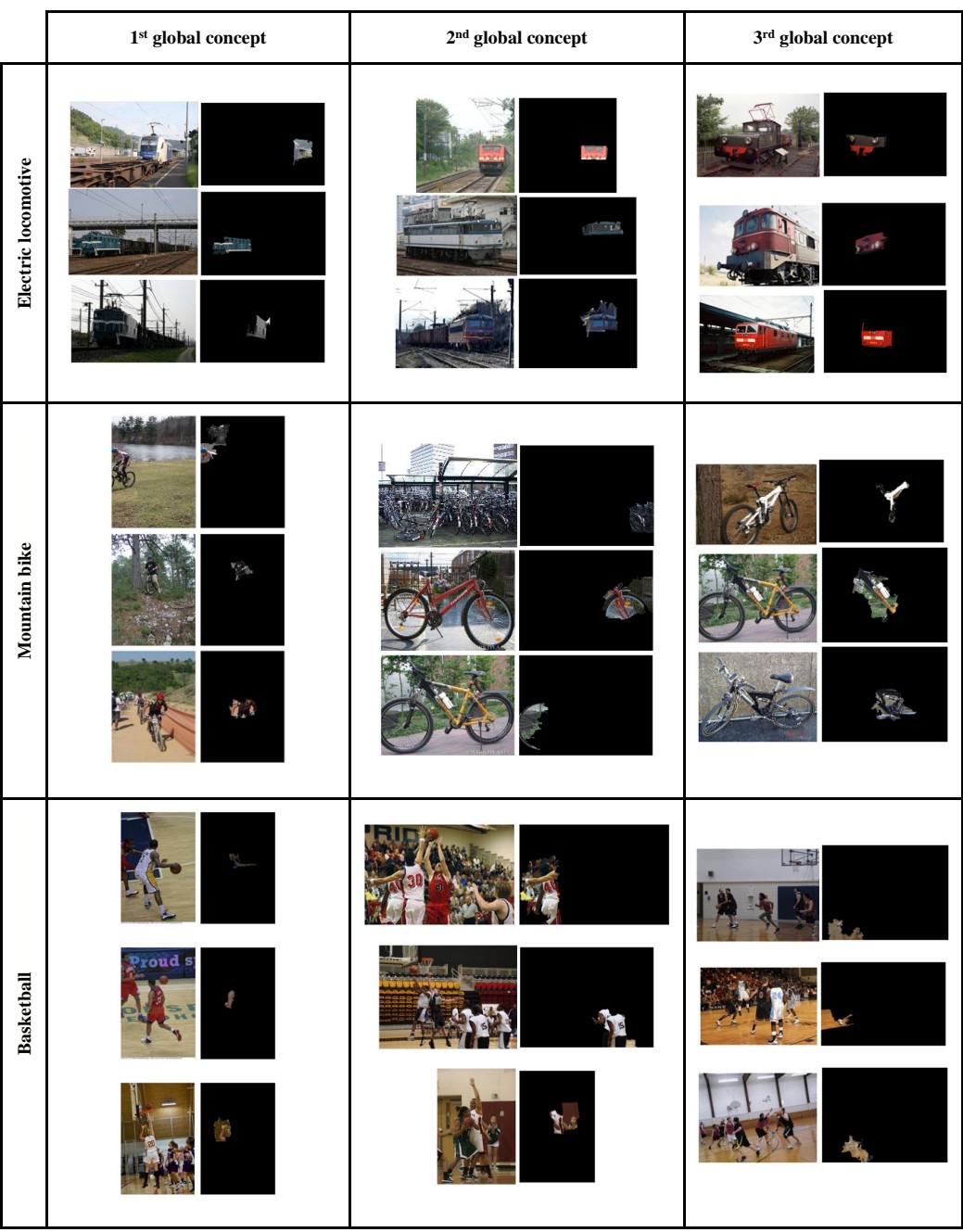

Supplementary Figure 15: Global concepts to explain one certain target class. For each class, the top three global concepts are indicated via the best three representative samples of each group.

| | Input sample | Super-pixels | Scores | Top local concepts |
|---|---|---|---|---|
| **Electric locomotive** | Electric locomotive predicted as others | | | |
| | others predicted as Electric locomotive | | | |
| **Mountain bike** | Mountain bike predicted as others | | | |
| | Others predicted as Mountain bike | | | |
| **Basketball** | Basketball predicted as others | | | |
| | others predicted as Basketball | | | |

Supplementary Figure 16: Explaining misclassification cases: two examples (if available) of FN and FP are represented for each target class, and the original image, the segmented version, the complete scoring map, and the scoring map of the top three local concepts are indicated for each example.

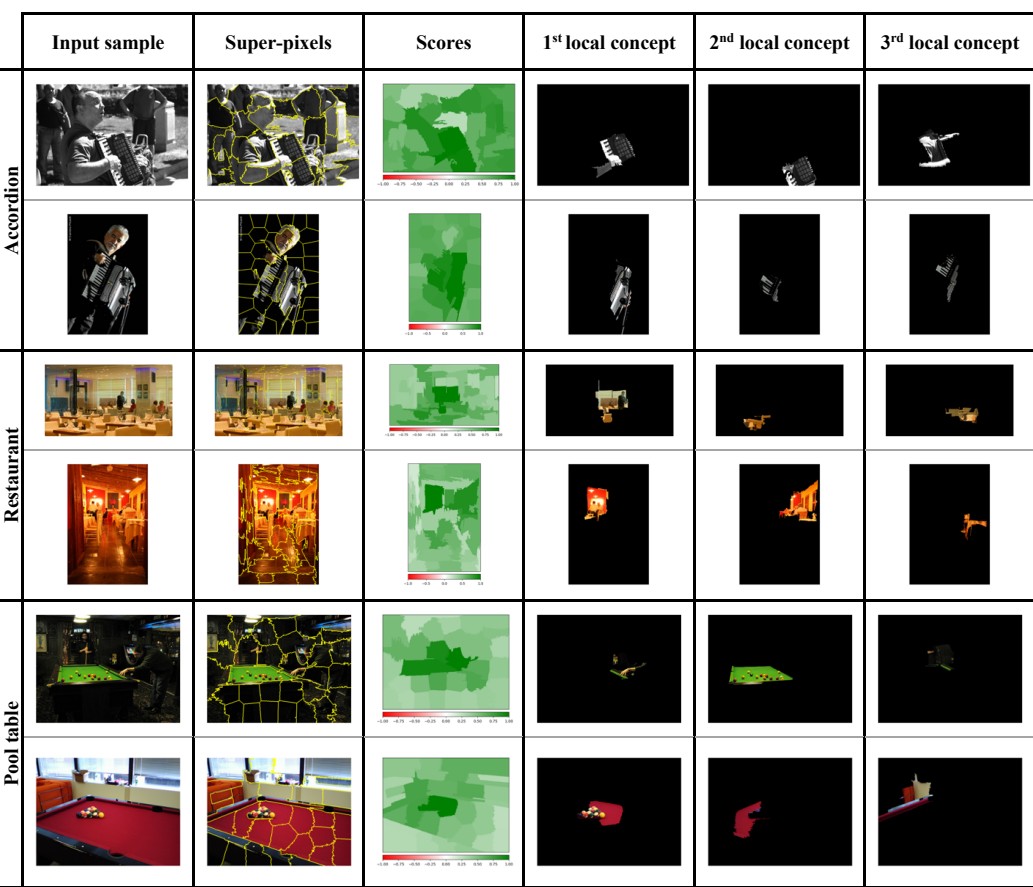

Supplementary Figure 17: Local concepts to explain individual images. For each class, there are two examples, displaying the original image, the segmented version, the complete scoring map, and the top three local concepts of each instance.

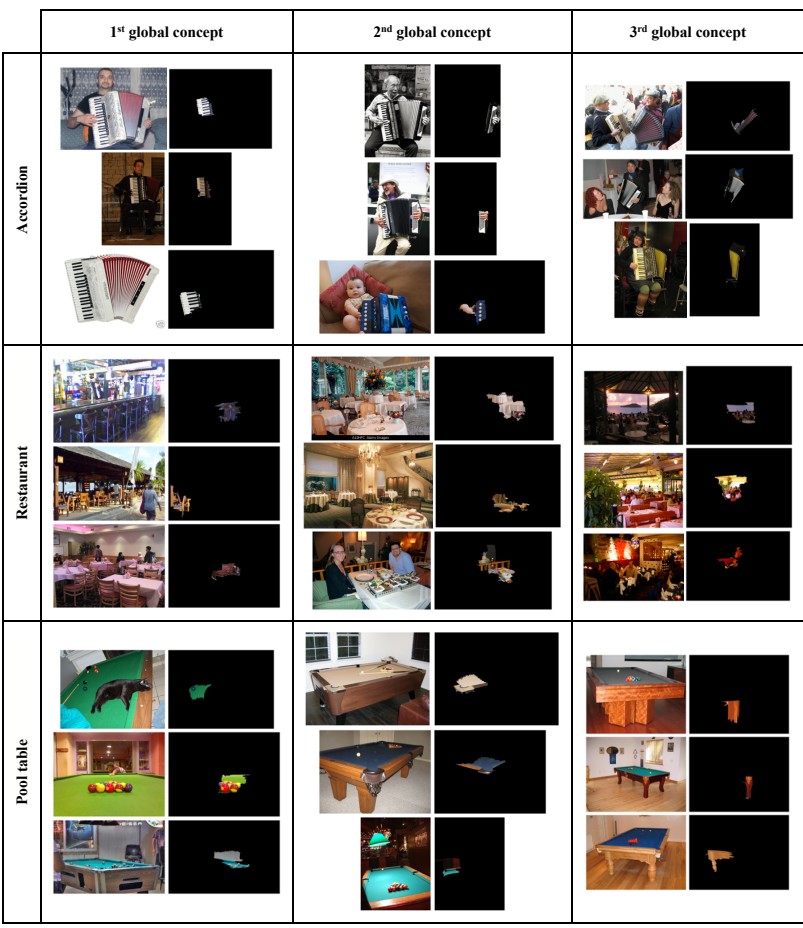

Supplementary Figure 18: Global concepts to explain one certain target class. For each class, the top three global concepts are indicated via the best three representative samples of each group.

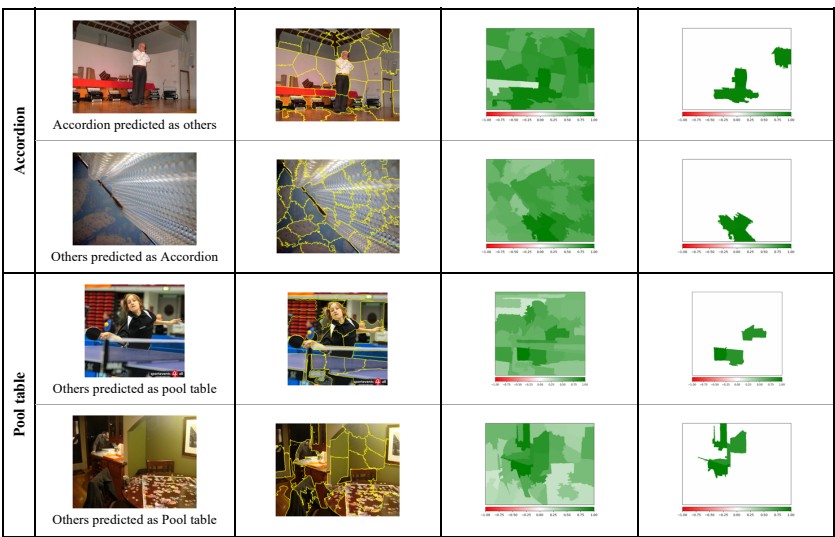

Supplementary Figure 19: Explaining misclassification cases: two examples (if available) of FN and FP are represented for each target class, and the original image, the segmented version, the complete scoring map, and the scoring map of the top three local concepts are indicated for each example.

