# OpenReview forum: "A Unified Concept-Based System for Local, Global, and Misclassification Explanations"
_ICLR.cc/2024/Conference — ICLR 2024 Conference Withdrawn Submission_

### Official Review · Reviewer_ZcJk · 2023-10-28

**Soundness:** 2 fair
**Presentation:** 2 fair
**Contribution:** 2 fair
**Rating:** 3
**Confidence:** 4

**Summary:**

This paper proposes a novel automated concept extraction and evaluation method that gives both local and global concepts to explain computer vision models. Their method takes inspiration from the field of Outlier detection, specifically using the softmax probability as an OOD detector, to fine-tune surrogate models that detect if concepts --image segments-- are important or not for a class. They evaluate their method against 4 attribution methods and show that their local explanation is competitive in terms of faithfulness and better in terms of completeness.

**Strengths:**

- The framing of concept importance as the "ood-ness" of a concept w.r.t to the class of interest is interesting
- The paper is overall clear
- The evaluation w.r.t to feature attribution methods that also use surrogate models to give local importance scores is comprehensive

**Weaknesses:**

- The contextualization of the paper in terms of prior concept-based methods is missing important and relevant methods.[1-2-3]
- Overall the evaluation is unconvincing. While the method is framed as an automated concept-based method, it does not compare itself to any other automated concept-based method [1-2], so it is hard to judge the proposed method
- Minor weakness but I would advise dropping the notion of  "ood" images in the methodology as here no samples are really ood in the true sense of the term (except arguably the segment but certainly not the image from other classes)

[1] Zhang et al. Invertible concept-based explanations for cnn models with non-negative concept activation vectors. In Proceedings
of the AAAI Conference on Artificial Intelligence, volume 35, pages 11682–11690, 2021.

[2] Fel et al. Craft: Concept recursive activation factorization for explainability. In Proceedings of the IEEE Conference on Computer Vision and Pattern Recognition (CVPR), 2023.

[3] Fel et al. A Holistic Approach to Unifying Automatic Concept Extraction and Concept Importance Estimation. arXiv. 2023

**Questions:**

- The definition of concept in the literature is usually of a direction in the latent space of a model. How would you define the notion of concept in your method?
- If I understand correctly, you fine-tune your surrogate model with images and their true label, is there a reason you do not use the output of the base model as a label instead?
- In the evaluation, talking about GradCAM and LRP you say "two attribution methods modified for the concept-based explanations" what does that mean?
- What is the motivation behind evaluating the generality of the concepts extracted by the proposed method? I would have argued that this is not necessarily something we want from an explainability method (which explains a specific model) to aim to be general so as to be able to explain different models at the same time.

---

### Official Review · Reviewer_2Hkv · 2023-10-30

**Soundness:** 2 fair
**Presentation:** 2 fair
**Contribution:** 1 poor
**Rating:** 3
**Confidence:** 5

**Summary:**

This paper centers on the subject of concept-based explanations in machine learning models. The authors introduce a unified framework designed to offer both local and global concept-based explanations, in addition to elucidating misclassifications. While the methodology proposed is straightforward, it is capable of generating cogent results.

**Strengths:**

Concept-based explanation in deep learning models constitutes a critical issue in the field of interpretable and explainable artificial intelligence.

The framework presented in the paper has the capability to furnish both local and global explanations, although it should be noted that pre-existing frameworks also possess this feature.

**Weaknesses:**

The adoption of a model tailored for binary classification tasks in UCBS appears to be a point of concern. This configuration prevents the concept-based system from contrasting the target class against a specific non-target class, leading to a loss of information. This approach diverges from the operation of deep learning (DL) classification models, which are typically trained to contrast multiple classes concurrently using a softmax function.

A lot of details are missing. Is the binary DNN classifier a separately trained network? Or is it using the encoder of the original network (the neural network it is explaining) and adding an additional binary classification head?

The authors assert that UCBS obviates the necessity for user-defined concepts and the application of external scoring methods (i.e., it eliminates “the need for providing user-defined concepts and applying external scoring methods”). However, the binary DNN classifier employed in their approach could arguably be considered an external tool. Furthermore, the paper lacks clarity on the distinct advantages of UCBS in comparison to other established methods like TCAV.

This leads to a closely related significant concern: the absence of pertinent baseline methods. There exists a substantial body of work on concept-based explanations for deep neural networks (DNNs), yet none of these are included as baselines in the paper. Consequently, it becomes unfeasible to assess UCBS's merits relative to the state-of-the-art. It's worth noting that the baselines employed in the paper, such as SHAP and LIME, are general methods and not specific to concept-based approaches.

Additionally, the authors characterize UCBS as an unsupervised learning method, a claim that appears misleading. UCBS relies on a segmentation neural network that is trained in a supervised manner. The labels used for training these segmentation networks essentially become the inherited concepts for UCBS. Therefore, this claim about the unsupervised nature of UCBS does not seem to be accurate.

**Questions:**

See comments above.

---

### Official Review · Reviewer_itZT · 2023-10-31

**Soundness:** 1 poor
**Presentation:** 1 poor
**Contribution:** 2 fair
**Rating:** 3
**Confidence:** 4

**Summary:**

This paper proposed UCBS, a new concept-based interpretation system to analyze the dataset of image classifiers locally and globally. The main idea is to train surrogate models classifying original/super-pixel images of the given class and randomly selected images, then leverage this model to rank and figure out the concepts. The paper showed that UCBS outperforms LIME, SHAP, LRP, and GradCAM in various metrics such as Insertion, Deletion, etc. I agree with the nature of the importance of this topic, but the method is too simple, and the motivation of the main idea is not described well.

**Strengths:**

- The idea is straightforward to follow and reproduce.
- The cost of the proposed method is lower than recently proposed saliency map-based explanation methods.
- UCBS outperformed the other baselines, such as LIME, SHAP, LRP, and GradCAM.

**Weaknesses:**

- Concerns on implementation
  - It is mentioned that the output of the convolution, maxpool, and avgpool with zero pixel area is zero. However, the color of the padded area in the figure is black, which means that the value of the padded area will not be zero after input normalization. This brings out the concerns that the implementation differs from the intentions described on page 3.
- Underestimated performance of baselines
  - The deletion score of LIME and GradCAM of the paper (https://arxiv.org/pdf/1806.07421.pdf) is 0.12, which is much lower than the numbers denoted in this paper.
- Presentations
  - The datasets $\mathcal{D^c}$, $\mathcal{D^c_{S}}$, and $\mathcal{D^c_{out}}$ are defined as sets of images in the notation section, while they seem sets of $(x, y)$ tuples in Eq (2).
  - I guess from Figure 1a that $y = 0$ for $\mathcal{D^c_{out}}$ and $y = 1$ for otherwise, but it is not denoted in the paper.
  - Notation $c$ is sometimes omitted without mention.
  - I guess the $Net_S$ should be a set of functions with the same numbers as class, $Net_S= ${$f^c_\theta$}.
  - The images in Figure 1 are too small.
- Absence of ablation study
  - I believe it is essential for this paper to show the ablation study, especially for concept learning. Specifically, we can replace the logit of surrogate model $f_\theta^{'c}$ as $IS(s_{i,j}) = f_\theta^{'c}(s_{i,j})$ with a pre-trained image classifier.

**Questions:**

- Is there any analysis that training using superpixel or segment helps the model to learn important regions more than background regions?
- Why there is $c$ in notation $f_\theta^{',c}$ while there is no $c$ in $f_\theta$?
- How about leveraging the image captioning models to see the concepts in the text?

---

### Official Review · Reviewer_u8Pp · 2023-10-31

**Soundness:** 2 fair
**Presentation:** 3 good
**Contribution:** 2 fair
**Rating:** 5
**Confidence:** 4

**Summary:**

The authors propose a novel framework called Unified Concept-Based System (UCBS) that provides concept-based explanations of Deep Neural Networks (DNNs).
Contrary to traditional eXplainable AI (XAI) methods that work on the level of single input features (pixels for image data), UCBS operates on the concept level providing more human-understandable explanations. Concepts are hereby defined as input feature groups (super-pixels) extracted from the dataset.

An essential feature of the UCBS framework is its ability to generate both local (instance-wise) and global (class-wise) explanations.
By comparing local and global explanations, UCBS enables a deeper understanding of misclassified predictions.
To compute local importance scores of concepts, surrogate models are employed. These surrogate models are fine-tuned versions of the original (explained) DNNs.

The authors illustrate the effectiveness of the UCBS approach through both qualitative and quantitative assessments, comparing it to established (local) XAI methods. Qualitatively, they provide examples of correct and incorrect predictions, showcasing how UCBS enhances interpretability. Quantitatively, the authors evaluate the model in terms of faithfulness, completeness, and generality."

**Strengths:**

The UCBS framework's capability to produce both local (instance-wise) and global (class-wise) explanations is a vital aspect.
Indeed, it seems to be very useful to combine both local and global explanations to gain an understanding of how individual predictions deviate from the typical case.

The methodology section of the UCBS approach is clearly described and straight-forward to understand.

The authors thoughtfully acknowledge limitations of UCBS, notably the dependency on super-pixels.

In the evaluation section, the qualitative examples provided are clear and effectively illustrate the practical utility of their approach.

**Weaknesses:**

The local component of UCBS appears to closely resemble traditional local XAI methods, such as SHAP and LIME, by focusing on super-pixel relevance scores.
However, it may be more insightful to explore the presence of global concepts within the local (prediction) context and their respective importance, akin to the methods introduced in CRP [1] and CRAFT [2].
Only the global UCBS component seems to be concept-based (by operating on the latent representations of the DNN).

In terms of the limitations of UCBS, a more comprehensive discussion regarding the number and relevance of hyper-parameters, particularly in the context of fine-tuning surrogate models, would provide a valuable addition. An ablation study investigating the influence of these hyper-parameters on the quantitative evaluation metrics would be of interest.

Concerning the quantitative evaluation, it's worth noting that the authors exclusively compare UCBS against local XAI methods. It would be beneficial to see a comparison against other concept-based XAI methods to provide a more comprehensive perspective. Additionally, the faithfulness evaluation curves are presented for only two methods and two samples in the appendix. Including the final curves used to calculate faithfulness metrics would enhance the overall value of the analysis."

[1] Achtibat, Reduan, et al. "From attribution maps to human-understandable explanations through Concept Relevance Propagation." Nature Machine Intelligence 5.9 (2023): 1006-1019.

[2] Fel, Thomas, et al. "Craft: Concept recursive activation factorization for explainability." Proceedings of the IEEE/CVF Conference on Computer Vision and Pattern Recognition. 2023.

**Questions:**

Can you motivate the choice of the generality metric? Would it be possible to compare UCBS here to some other concept-based XAI method?

It would be also interesting to compare your (local) component against other local XAI approaches (LRP, SHAP, etc.) in terms of computational complexity, as it seems to be only based on few DNN forward passes (one for each super-pixel).

---

### Official Review · Reviewer_DJR7 · 2023-10-31

**Soundness:** 2 fair
**Presentation:** 3 good
**Contribution:** 3 good
**Rating:** 6
**Confidence:** 4

**Summary:**

The authors propose a unified framework for generating both local and global concept based explanations. The proposed framework considers a surrogate model trained with superpixels of an image for generating concepts, which are used then used to extract local and global explanations.

**Strengths:**

- The proposed approach is innovative and can be used to generate and score the relevance of concepts
- The illustrations of local explanations are well presented, nice framework and illustration figures
- The paper is well written and easy to follow

**Weaknesses:**

- The proposed approach is certainly not the first to explore the extraction of local and global explanations, please refer to [1, 2].
- What is the motivation for having the out-of-distribution dataset to train the surrogate model?
- The proposed approach only captures the concepts that can be described as a part of an image. Is it possible to capture concepts like lighting that are expressed in an entire image?
- I'm unsure of how to interpret global explanations. Can you please provide more discussion for the same?
- It is a bit unclear how the scores in Table 1 are computed, given the UCBS is fine-tuned on superpixel images; the scores like insertion, deletion, and faithfulness are expected to be high for UCBS, given they are on the manifold, while in case of LIME, SHAPE, GradCAM, and LRP the concepts are off-manifold resulting in lower score. Could you please justify this?



[1] Kori, A., Glocker, B. and Toni, F., 2022. GLANCE: Global to Local Architecture-Neutral Concept-based Explanations. _arXiv preprint arXiv:2207.01917_.

[2] Setzu, M., Guidotti, R., Monreale, A., Turini, F., Pedreschi, D. and Giannotti, F., 2021. Glocalx-from local to global explanations of black box ai models. _Artificial Intelligence_, _294_, p.103457.

**Questions:**

Please refer to weakness section

---

> ### Author Response · Authors · 2023-11-18
> **Reply to Reviewer DJR7**
>
> We are grateful for your thorough review and insightful suggestions.  Below, we address your concerns mentioned in the Weaknesses.
>
> 1. The proposed approach is certainly not the first to explore the extraction of local and global explanations, please refer to [1, 2].
>
> Thanks for your valuable mention. In our paper, we initially stated that "UCBS is, to the best of our knowledge, one of the first unsupervised unified concept extraction frameworks." This claim is based on three important factors: the unified nature encompassing local, global, and misclassification explanations, the focus on concept-based extraction, and the unsupervised nature in terms of explanation labels. We meant to emphasize that there are few concentrated works in these areas targeting vision-based applications, making UCBS possibly one of the pioneering methods. However, in order to avoid potential confusion, we decided to remove this statement.
>
> More clarification: Ref [2] has primarily been designed for classification tasks on tabular data, and we believe that its applicability to pixel-based data may be limited due to its rule-based nature. Furthermore, Ref [1] offers global explanations in terms of graphs and local explanations in terms of feature importance scores, not concepts.
>
>
> 2. What is the motivation for having the out-of-distribution dataset to train the surrogate model?
>
> Surrogate models are indeed binary classifiers that try to fully comprehend the target of interest.  In order to accomplish this, their initial task is to differentiate the target's objects from the non-target samples. Subsequently, they should be able to discriminate the target's concepts and assign them scores based on their significance. The out-of-distribution samples (the non-target samples), which are known relative to the distribution of the target samples, are crucial for achieving the first objective. These samples, in companion with the target samples and the super-pixelated images, are utilized to teach the surrogate network a better representation of the target’s objects as well as the target’s concepts. In order to further clarify this comment, we revised the 5-th paragraph of the introduction as follows:
> In an analogous case, we engage surrogate explainer networks and equip them with auxiliary datasets to purposely concentrate on the target of interest, rather than a multi-class classification approach. That is, the surrogate networks try to comprehensively learn one target class against the non-target samples. This allows them to thoroughly learn the target's objects as well as their informative concepts.
>
> 3. The proposed approach only captures the concepts that can be described as a part of an image. Is it possible to capture concepts like lighting that are expressed in an entire image?
>
> No, the current version of the method is not able to detect abstract and complex concepts like brightness or daylight, which are challenging concepts necessitating additional pre/post-processing tasks. We mentioned this limitation in section 5 (Discussion, limitations, and future works) as follows:
>
> However, it is important to acknowledge that the limitations and drawbacks of superpixels are inherited by the UCBS, which may potentially result in the creation of meaningless concepts. Moreover, the automatic extraction of abstract and complex concepts may be challenging or necessitate additional post-processing. Addressing these challenges presents an intriguing direction for future research, particularly in identifying and surrounding the most influential concepts within bounding boxes, which could help mitigate the issues.
>
> In addition, in appendix section C.5 (More discussion on limitations: Imperfectly super-pixelated inputs and Dataset biases), we further discussed that this limitation refers to incorporating super-pixelating methods:
>
> Moreover, super-pixelating methods typically offer no solution for tracking entirely or partially abstract concepts such as brightness, daylight, and sun angle in sunrise or sunset images.